# Inorganic Arsenic Exposure Decreases Muscle Mass and Enhances Denervation-Induced Muscle Atrophy in Mice

**DOI:** 10.3390/molecules25133057

**Published:** 2020-07-04

**Authors:** Chang-Mu Chen, Min-Ni Chung, Chen-Yuan Chiu, Shing-Hwa Liu, Kuo-Cheng Lan

**Affiliations:** 1Department of Surgery, College of Medicine and Hospital, National Taiwan University, Taipei 100, Taiwan; cmchen10@ntuh.gov.tw; 2Institute of Toxicology, College of Medicine, National Taiwan University, Taipei 100, Taiwan; r03447001@ntu.edu.tw; 3Department of Botanicals, Medical and Pharmaceutical Industry Technology and Development Center, New Taipei City 248, Taiwan; cychiu@pitdc.org.tw; 4Department of Medical Research, China Medical University Hospital, China Medical University, Taichung 404, Taiwan; 5Department of Pediatrics, College of Medicine, National Taiwan University & Hospital, Taipei 100, Taiwan; 6Department of Emergency Medicine, Tri-Service General Hospital, National Defense Medical Center, Taipei 114, Taiwan

**Keywords:** arsenic, skeletal muscle, denervation, atrophy

## Abstract

Arsenic is a toxic metalloid. Infants with a low birth-weight have been observed in areas with high-level arsenic in drinking water ranging from 463 to 1025 μg/L. A distal muscular atrophy side effect has been observed in acute promyelocytic leukemia patients treated with arsenic trioxide (As_2_O_3_) for therapy. The potential of As_2_O_3_ on muscle atrophy remains to be clarified. In this study, the myoatrophic effect of arsenic was evaluated in normal mice and sciatic nerve denervated mice exposed with or without As_2_O_3_ (0.05 and 0.5 ppm) in drinking water for 4 weeks. We found that both 0.05 and 0.5 ppm As_2_O_3_ increased the fasting plasma glucose level; but only 0.5 ppm arsenic exposure significantly decreased muscle mass, muscle endurance, and cross-sectional area of muscle fibers, and increased muscle Atrogin-1 protein expression in the normal mice. Both 0.05 and 0.5 ppm As_2_O_3_ also significantly enhanced the inhibitory effects on muscle endurance, muscle mass, and cross-sectional area of muscle fibers, and increased the effect on muscle Atrogin-1 protein expression in the denervated mice. These in vivo results suggest that inorganic arsenic at doses relevant to humans may possess myoatrophic potential.

## 1. Introduction

The sources of inorganic arsenic exposed to humans include contaminated drinking water, contaminated food, industrial processes, contaminated water used for food preparation and crop irrigation, and tobacco smoking [1]. The long-term exposure to arsenic from these sources can cause cancer, skin lesions, cardiovascular disease, and diabetes [1]. Arsenic in the drinking water at levels above 10 μg/L, a WHO provisional guideline value, has been estimated to be at least 140 million people in 50 countries [1]. A previous study has reported that a higher risk for preterm birth, stillbirth, miscarriage, and low birth-weight in offspring is found in women who drink contaminated water containing from 463 to 1025 μg/L arsenic [2]. The side effects such as pleura-pericardial effusion, polyneuropathy, weight gain, and distal muscular atrophy have also been observed in patients administered with As_2_O_3_ that acts as a clinical agent against acute promyelocytic leukemia (APL) [3,4].

The characterization of muscle atrophy includes a reduction in muscle protein content, muscle fiber cross-sectional area, myonuclear number, and muscle strength, and an increase in fatigability. Muscle loss can cause abnormal physical abilities and metabolic alterations such as osteoporosis and insulin resistance [5,6,7]. Arsenic trioxide (As_2_O_3_) has been shown to suppress myogenic differentiation via an inhibition of the Akt signaling pathway in cultured myoblasts and retard muscle regeneration in a glycerol-induced myopathy mouse model [8]. Recently, submicromolar-concentration As_2_O_3_ has also been shown to be capable of inducing myotube atrophy in a myoblast cell model [9]. However, the in vivo effect of As_2_O_3_ on skeletal muscle atrophy remains to be clarified.

Arsenic levels of 671 ± 149 (470–897) ppb have been detected in the underground water in endemic arsenic contamination areas of Taiwan, and trivalent arsenite is a predominant arsenic species [10]. As_2_O_3_, which can form arsenite in an alkaline solution, is the most commonly produced form of arsenic [11,12]. In this study, we hypothesized that As_2_O_3_ is an environmental risk factor for muscle atrophy. We investigated the effects of As_2_O_3_ at levels relevant to human exposure from drinking water on skeletal muscle atrophy in normal mice and denervated mice. To avoid the neural interference, a sciatic nerve denervation-induced muscle atrophy mouse model was used.

## 2. Results and Discussion

Exposure of inorganic arsenic can induce neurotoxicity in experimental animals and humans [13,14,15]. Denervation is known to damage the balance between protein synthesis and protein degradation in skeletal muscles, causing muscle atrophy [16]. In this study, we used a sciatic nerve resection model to avoid the influence of As_2_O_3_-induced peripheral neurotoxicity in skeletal muscles. We investigated the in vivo effects of As_2_O_3_ (0.05 and 0.5 ppm) exposed from drinking water for 4 weeks on muscle growth and function during both normal and denervation conditions. 

We observed that As_2_O_3_ (0.05 and 0.5 ppm) exposure for 4 weeks did not change body weights in the normal condition, but decreased body weights in the denervation condition (Figure 1A). As_2_O_3_ exposure did not change the animal behavior during experiment. As_2_O_3_ (0.05 and 0.5 ppm) exposure could also increase the fasting blood glucose in mice with or without denervation (Figure 1B). The increased effects of arsenic on blood glucose are consistent with the previous findings. A case-cohort study by James et al. (2013) [17] has shown that an increased type 2 diabetes risk is associated with the lifetime exposure to low-concentration arsenic from drinking water. Huang et al. (2015) [18] have shown that As_2_O_3_ at doses relevant to human exposure from drinking water is capable of impairing glucose and insulin tolerance and inducing insulin resistance in an estrogen-deficient female mouse model.

An association has been shown between high-level arsenic exposure from drinking water in certain areas and increased low birth-weight rate and impaired muscle regenerative capacity [2]. As_2_O_3_ has also been found to suppress myogenic differentiation and muscle regeneration after muscle injury by glycerol in mice [8]. Jensen et al. (2007) [19] have suggested an association between altered muscle fiber composition and size and developing type 2 diabetes in young men with a low birth-weight. Several studies have indicated that diabetes is a risk factor for age-associated progressive and generalized loss of skeletal muscle mass and function (sarcopenia) [20,21,22]. Therefore, we next investigated the effects of arsenic exposure on muscle atrophy in a devervated mouse model.

The gross morphology of the soleus, gastrocnemius, and tibialis anterior muscles showed a marked reduction in muscle size in the denervated mice with or without As_2_O_3_ treatment (Figure 2A). The muscle masses of soleus, gastrocnemius, and tibialis anterior muscles in the denervated mice with As_2_O_3_ (0.05 and 0.5 ppm) treatment were significantly less than that in the denervated mice alone (Figure 2B). The muscle masses of soleus, gastrocnemius, and tibialis anterior muscles in the normal mice with As_2_O_3_ 0.5 ppm, but not 0.05 ppm, treatment were significantly decreased as compared to the control mice (Figure 2B).

To evaluate muscle function, we further observed the endurance of skeletal muscles with muscle fatigue using the rotarod performance test. As shown in Figure 3, the latency to fall on the rotarod apparatus was slightly but significantly shortened in the As_2_O_3_ (0.5 ppm)-exposed normal mice, and was markedly and significantly shortened in the denervated mice with or without As_2_O_3_ treatment. The latency to fall in the denervated mice with As_2_O_3_ (0.05 and 0.5 ppm) treatment was significantly shorter than that in the denervated mice alone (Figure 3). Especially, the denervated mice with As_2_O_3_ (0.05 and 0.5 ppm) treatment showed extreme fatigue after 5 min of the test (Figure 3). 

A decrease in the fiber cross-sectional area occurs in response to the loss of muscle mass which represents muscle weakness [23]. We next observed and calculated the cross-sectional area of muscle fibers. The soleus muscle fiber atrophy was observed by hematoxylin and eosin staining (Figure 4A) and cross-sectional area reduction (Figure 4B), which performed a leftward shift in the representative frequency distribution histogram (Figure 4C) in the As_2_O_3_ (0.5 ppm)-treated normal mice and the denervated mice with or without As_2_O_3_ (0.05 and 0.5 ppm) treatment. The As_2_O_3_ (0.05 and 0.5 ppm) exposure in combination with denervation treatment significantly enhanced the reduction in the cross-sectional area, which performed a leftward shift in the representative frequency distribution histogram in soleus muscle as compared to both denervation alone and As_2_O_3_ alone (Figure 4). 

The large rotator cuff tears often accompany muscle atrophy and fatty infiltration even after the tendon is repaired that is correlated with a poor functional outcome [24]. Fatty infiltration, which adipocytes home below the fascia of the muscle, is thought to delay or hinder the muscle repair process after injury. Fatty infiltration has been suggested to elicit inflammatory responses inside the muscles in rotator cuff lesions, resulting in interferences with glucose metabolism [25]. Muscle atrophy and fatty infiltration have also been observed in the chronic denervation of muscles [26]. In the present study, the muscle morphology in mice with As_2_O_3_ exposure or denervation, or a combination of both, seemed to reveal more vacant spaces in soleus muscles compared to the normal control mice. This may be associated with the fatty infiltration. The detailed effect and mechanism of fatty infiltration in muscles of the As_2_O_3_-exposed mice need to be clarified in the future.

Atrogin-1/MAFbx, one of the muscle-specific ubiquitin ligases is involved in the increased protein degradation [27]. We also found that the protein expression of the muscle atrophy marker Atrogin-1 was significantly increased in soleus muscles from both the As_2_O_3_ (0.5 ppm)-treated normal mice and the denervated mice with or without As_2_O_3_ treatment (Figure 5). The treatment of denervation in combination with As_2_O_3_ (0.05 and 0.5 ppm) exposure significantly enhanced the increase in Atrogin-1 protein expression in soleus muscles as compared to both denervation alone and As_2_O_3_ alone (Figure 5). These results indicate that arsenic has a potential to induce muscle atrophy in vivo.

Several signaling pathways are known to be involved in muscle atrophy, and the signaling molecules such as Akt, AMPK, FoxOs, NFκB, and c-Jun N-terminal kinase have been suggested to be important regulators for the expressions of atrogenes Atrogin-1 and MuRF1 [27,28,29,30]. The negative regulation of FoxO-upregulated atrogin-1 and MuRF1 by the activated Akt signaling has been found to block muscle atrophy [27]. Chiu et al. (2020) [9] have recently found that submicromolar-concentration As_2_O_3_ triggered Atrogin1 and MuRF1 protein expressions and myotube atrophy in vitro via an inhibition of the Akt-related signaling pathway. In the present study, we further found that low-dose As_2_O_3_ exposure for 4 weeks significantly enhanced the muscle atrophy and triggered muscle Atrogin-1 expression in a denervation-induced muscle atrophy mouse model. Nevertheless, MacDonald et al. (2014) [16] have suggested that the pathophysiological mechanisms of disuse- and denervation-induced muscle atrophy were fundamentally different. They found that disuse-induced atrophy could be prevented by myostatin inhibition, but denervation-induced atrophy was independent from the activation of Akt and its downstream mTOR and could not be reversed by myostatin inhibition. The muscle atrophy caused by different conditions or models may be distinct from each other and have different underlying pathophysiological mechanisms. The detailed molecular mechanism in As_2_O_3_-enhanced muscle atrophy in denervated mice remains subject to further investigation.

Exposure to inorganic arsenic can induce neurotoxicity in experimental animals and humans [13,14,15]. In the present study, we found that 0.5 ppm As_2_O_3_ could induce atrophy of muscle fibers in normal mice without or with denervation. Nevertheless, human neurodegenerative disorders are known to induce muscle atrophy, such as spinal and bulbar muscular atrophy [31,32]. The role of neurotoxicity of androgens (e.g., testosterone and its derivative dihydrotestosterone) in human spinal and bulbar muscular atrophy has been suggested [32]. Therefore, it may not be possible to rule out the role of neurotoxicity of arsenic in 0.5 ppm As_2_O_3_-induced muscle atrophy in normal mice without denervation. On the other hand, there is the subclinical muscle atrophy in humans. Rocchi et al. (2011) [31] have suggested the subclinical involvement of the autonomic dysfunction in human spinobulbar muscular atrophy. The subclinical features have been characterized in an autosomal dominant spinal muscular atrophy caused by the heterozygous variants in the BICD cargo adapter 2 (BICD2) [33]. Fischmann et al. (2012) [34] have also shown that the subclinical changes in patients with oculopharyngeal muscular dystrophy can be detected by conventional and quantitative muscle magnetic resonance imaging (MRI). However, the subclinical muscle atrophy induced by arsenic exposure in animals or humans still requires further investigation.

## 3. Materials and Methods

### 3.1. Animal Experiment

Male 6-week-old Institute of Cancer Research (ICR; CD1) mice were provided by the Animal Center of the College of Medicine, National Taiwan University (Taipei, Taiwan). The ethical review committee of the College of Medicine, National Taiwan University approved this animal study. The animal experiments were performed according to the regulations of Taiwan and National Institutes of Health (NIH) guidelines on the care and welfare of laboratory animals. The animals were humanely treated with regard to the alleviation of suffering. Mice were housed in the standard mouse microisolator cages with aspen chip bedding under controlled conditions (22 ± 2 ℃, 12 h light/dark cycles). All mice were given a standard chow diet (LabDiet #5053) and water ad libitum. There was a one-week acclimation for all mice. Mice with or without muscle denervation were exposed to As_2_O_3_ 0.05 or 0.5 ppm through drinking water for 4 consecutive weeks (n = 6 mice/group). The selection of compound doses and periods of study was according to our previous studies [8,18] and our preliminary test. The As_2_O_3_-containing drinking water was freshly prepared every 3 days to prevent or minimize As_2_O_3_ oxidation.

### 3.2. Muscle Denervation Model

One hindlimb of mice was denervated by the surgical removal of at least 5 mm of sciatic nerve under anesthesia with the inhalational application of a mixture gas of isoflurane and oxygen, as described previously [16]. Sham control mice were operated in the same way without sciatic nerve removal. The surgical staples were used to close wounds. After the denervation surgery, mice were exposed to As_2_O_3_ in drinking water for 4 weeks. Muscle strength was assessed after As_2_O_3_ exposure. At the end of experiments, anesthetized animals were humanely sacrificed. Skeletal muscles (soleus, tibialis anterior (TA), and gastrocnemius (GAS) muscle) were harvested and weighed. Portions of these muscles were frozen at −80 ℃. Other portions were fixed over-night in 4% paraformaldehyde at room temperature for following histological analysis.

### 3.3. Detection of Muscle Fatigue Task

The detection of muscle fatigue task was determined as previously described [35]. Briefly, mice were training on a rotarod apparatus prior to the fatigue task (Ugo Basile, Varese, Italy). In the training trial, each mouse was placed on the rotarod at a constant speed of 13 rpm for 15 min. After 15 min of rest, each mouse was placed back on the rotarod at speeds ramping from 13 to 25 rpm in 180 s for 15 min. The next day, mice completed the muscle fatigue task on the rotarod at speeds ramping from 13 to 25 rpm in 180 s and maintained at 25 rpm for a total duration of 30 min run time with 10 min rest intervals. The muscle fatigue in mice could be measured while the mouse fell off four times within one min of each fall, and then the task terminated. The latency to fall off the rotarod was recorded for each individual mouse.

### 3.4. Histological and Immunohistochemical Analysis

Fixed skeletal muscles were paraffin-embedded, and prepared as 4-μm thick sections for hematoxylin and eosin staining and immunohistochemistry. For immunohistochemistry, the sections were blocked with 5% bovine serum albumin for 1 h under room temperature, and then incubated with a primary antibody for Atrogin-1 (Abcam, Cambridge, UK) overnight at 4 ℃. The SuperPicture horseradish peroxidase polymer conjugate (Invitrogen, Carlsbad, CA, USA) was used to detect the primary antibody. The sections were subsequently counterstained with hematoxylin. The quantification of myofiber cross-sectional areas and the relative optical density of immunohistochemical images were analyzed by ImageJ analysis software (National Institutes of Health) in five random visual fields of each section.

### 3.5. Statistical Analysis

Results are shown as the mean ± SEM. The statistical significance of differences was analyzed by a one-way analysis of variance (ANOVA) and an unpaired two-tailed Student’s t-test with *p* < 0.05. The GraphPad Prism 6 software was used for the statistical analysis. 

## 4. Conclusions

In several parts of the world, high-level inorganic arsenic can be found in groundwater used as drinking water. Levels of arsenic in drinking water as high as 4700 μg/L have been detected, and levels of more than 300 μg/L are common [36,37]. Arsenic is associated with low birth-weight infants in areas with high-level arsenic in drinking water [2]. Distal limb weakness/atrophy is evident in arsenicosis patients from areas of groundwater arsenic contamination in West Bengal, India [38]. The present findings demonstrate for the first time that As_2_O_3_ at doses relevant to human exposure can induce skeletal muscle atrophy in vivo. We found that As_2_O_3_ (0.05 and 0.5 ppm) exposure for 4 weeks enhanced muscle atrophy and muscle weakness in mice with sciatic nerve denervation. Nonetheless, As_2_O_3_ at a dose of 0.5 ppm by itself significantly induced muscle atrophy in the normal mice. These in vivo results suggest that inorganic arsenic may be an environmental risk factor for muscle atrophy.

## Figures and Tables

**Figure 1 molecules-25-03057-f001:**
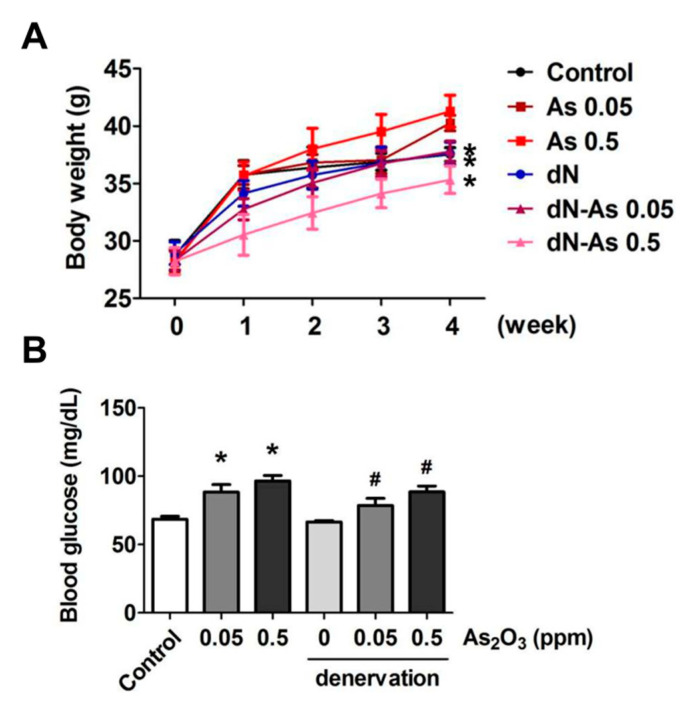
Effects of As_2_O_3_ on the body weight and blood glucose in mice with or without denervation. As_2_O_3_ (0.05 and 0.5 ppm) exposure from drinking water for 4 weeks in normal mice or denervated mice. (**A**). The changes in body weights were shown. (**B**). The blood glucose levels in mice were shown. Data are presented as the mean ± SEM (*n* = 6 of each group). * *P* < 0.05 compared to the control group; # *P* < 0.05 compared to the denervation group.

**Figure 2 molecules-25-03057-f002:**
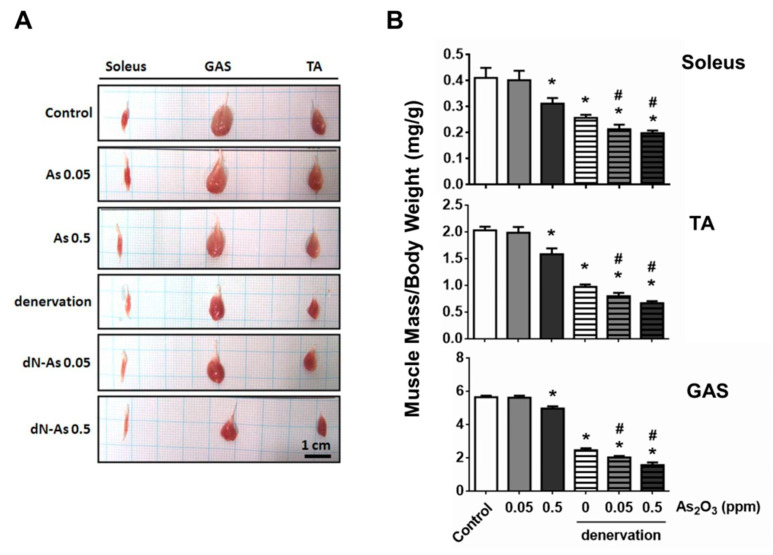
Effects of As_2_O_3_ on the muscle mass in mice with or without denervation. As_2_O_3_ (0.05 and 0.5 ppm) exposure from drinking water for 4 weeks in normal mice or denervated mice. (**A**). The gross morphology of each group in soleus, tibialis anterior (TA), and gastrocnemius (GAS) muscles were shown. Scale bar = 1 cm. (**B**). The average muscle mass of soleus, tibialis anterior, and gastrocnemius muscles were shown. Data are presented as the mean ± SEM (*n* = 6 of each group). * *P* < 0.05 compared to control group; # *P* < 0.05 compared to denervation group.

**Figure 3 molecules-25-03057-f003:**
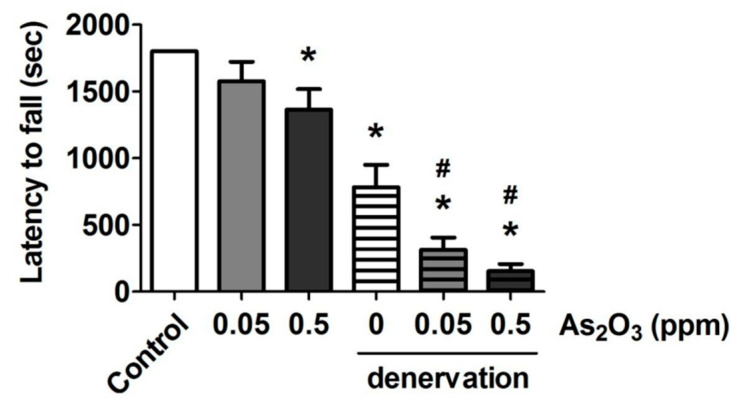
Effects of As_2_O_3_ on the muscle endurance in mice with or without denervation. As_2_O_3_ (0.05 and 0.5 ppm) exposure from drinking water for 4 weeks in normal mice or denervated mice. Muscle endurance was determined as latency to fall in the muscle fatigue task with rotarod. Data are presented as the mean ± SEM (*n* = 6 of each group). * *P* < 0.05 compared to the control group; # *P* < 0.05 compared to the denervation group.

**Figure 4 molecules-25-03057-f004:**
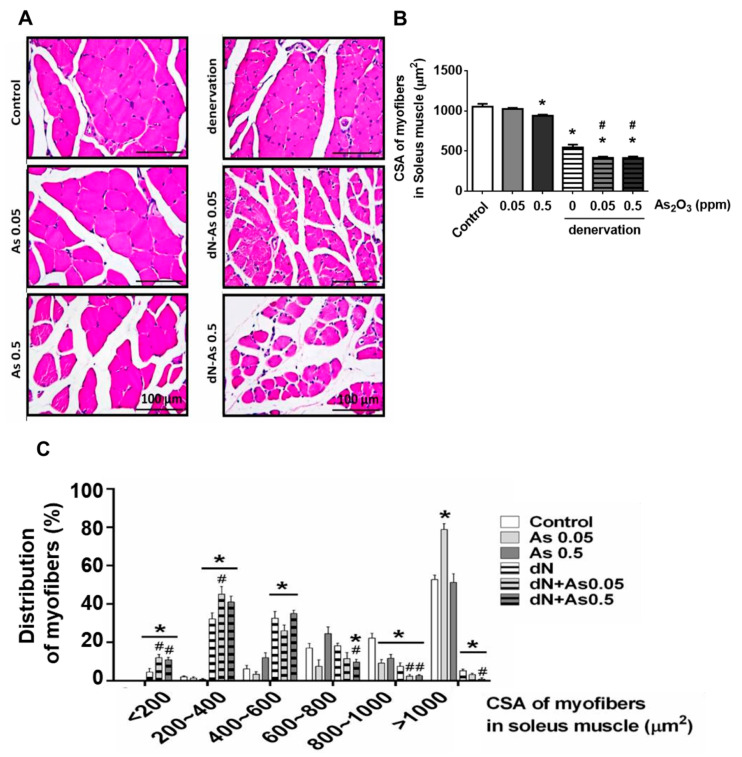
Effects of As_2_O_3_ on muscle fiber cross-sectional area (CSA) in mice with or without denervation. As_2_O_3_ (0.05 and 0.5 ppm) exposure from drinking water for 4 weeks in normal mice or denervated mice. The representative hematoxylin and eosin -stained muscle sections (**A**), the average cross-sectional areas (CSA) of myofibers (**B**), and the frequency distribution of myofibers (**C**) in soleus muscles were shown. Scale bar = 100 μm. Data are presented as the mean ± SEM (*n* = 6 of each group). * *P* < 0.05 compared to the control group; # *P* < 0.05 compared to the denervation group.

**Figure 5 molecules-25-03057-f005:**
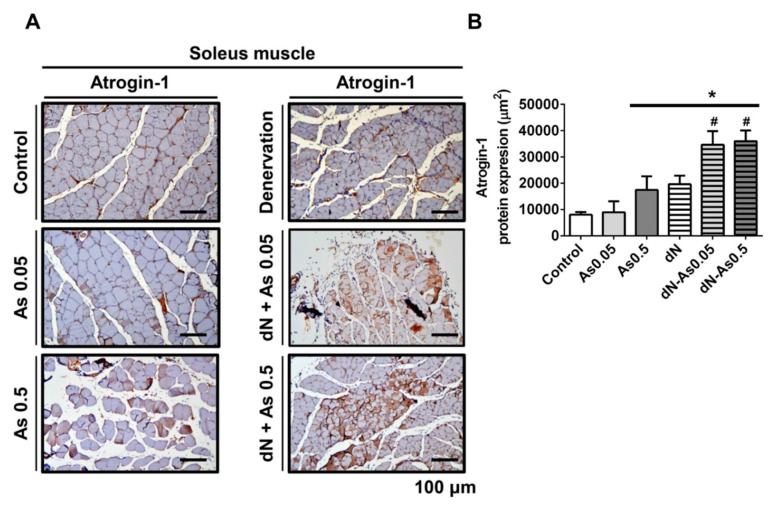
Effects of As_2_O_3_ on the immunohistochemical change of Atrogin-1 expression in soleus muscles of mice with or without denervation. As_2_O_3_ (0.05 and 0.5 ppm) exposure from drinking water for 4 weeks in normal mice or denervated mice. (**A**). The representative immunohistochemical images for the expression of Atrogin-1 in the soleus muscles isolated from As_2_O_3_-treated mice with or without denervation were shown. Scale bar = 100 μm. (**B**). The quantitative analysis of immunohistochemistry was shown. Data are presented as the mean ± SEM (*n* = 6 of each group). * *P* < 0.05 compared to the control group; # *P* < 0.05 compared to the denervation group.

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
