# Peer review of "Inorganic Arsenic Exposure Decreases Muscle Mass and Enhances Denervation-Induced Muscle Atrophy in Mice"

_molecules, 2020, doi:10.3390/molecules25133057_

Round 1

Reviewer 1 Report

In this manuscript, the authors did in vivo study to investigate the potential myoatrophic effect of arsenic in normal mice and sciatic  nerve denervated mice exposed with or without As2O3 (0.05 and 0.5 ppm) in drinking water for 4 weeks. They found that doses of As2O3 (0.05 and 0.5 ppm) increased the fasting plasma glucose level; but only 0.5 ppm arsenic exposure significantly decreased muscle mass, muscle endurance, and  cross-sectional area of muscle fibres, and increased muscle Atrogin-1 protein expression in normal mice. From a toxicological perspective, this is an interesting study that provide more insight into neurotoxicity and behaviour performance under arsenic environment. This manuscript is well written, organized and sounds with a good standard of English language. The methodologies are appropriate and aligned with the proposed objectives. The message from this manuscript is quite meaningful. I have a few issues need to be addressed as described below:

  1. How compound doses and period of study were selected?
  2. As2O3 -treated mice showed aggressive behavioural components?
  3. The resolution for some figures is not good, especially, fig4. It was so hard to check fig4C, the fig size should be increased.

Author Response

Reviewer#1

In this manuscript, the authors did in vivo study to investigate the potential myoatrophic effect of arsenic in normal mice and sciatic nerve denervated mice exposed with or without As2O3 (0.05 and 0.5 ppm) in drinking water for 4 weeks. They found that doses of As2O3 (0.05 and 0.5 ppm) increased the fasting plasma glucose level; but only 0.5 ppm arsenic exposure significantly decreased muscle mass, muscle endurance, and cross-sectional area of muscle fibres, and increased muscle Atrogin-1 protein expression in normal mice. From a toxicological perspective, this is an interesting study that provide more insight into neurotoxicity and behaviour performance under arsenic environment. This manuscript is well written, organized and sounds with a good standard of English language. The methodologies are appropriate and aligned with the proposed objectives. The message from this manuscript is quite meaningful. I have a few issues need to be addressed as described below:

  1. How compound doses and period of study were selected?

Response: We appreciate the reviewer's comments. The selection for compound doses and period of study is according to our previous studies [Yen et al., 2010; Huang et al., 2015 (references of number 8 and 17 of this manuscript)] and our preliminary test. We have also added this description in the Methods of this revised manuscript.

  1. As2O3 -treated mice showed aggressive behavioural components?

Response: We appreciate the reviewer's comments. In this study, exposure of As2O3 at levels relevant to human exposure from drinking water (0.05 and 0.5 ppm) in mice did not change the animal behavior during experiment. We have also added this description in the Results of this revised manuscript.

  1. The resolution for some figures is not good, especially, fig4. It was so hard to check fig4C, the fig size should be increased.

Response: We appreciate the reviewer's comments. We have checked and revised the figures, especially Fig. 4, in this revised manuscript according to the suggestion of reviewer.

Reviewer 2 Report

In this paper, authors investigated the toxic effect of As2O3 on muscle. They found that both 0.05 and 0.5 ppm As2O3 increased the fasting plasma glucose level; but only 0.5 ppm arsenic exposure significantly decreased muscle mass, muscle endurance, and cross-sectional area of muscle fibers. Their conclusion is that inorganic arsenic at doses relevant to human may possess the myoatrophic potential.

Regarding the presence or absence of muscular atrophy of arsenic, there is no previous report and it is a new focus. Studies have also carefully considered the cause using atrogin-1. The reviwer has only some minor concerns.

Although past reports have shown that inorganic arsenic has little effect on muscle, it has a strong effect on peripheral nerves in human (Int J Mol Sci. 2019 Jul 11;20(14):3418. doi: 10.3390/ijms20143418). In histological analysis, are there grouping atrophy of muscle fibers in 0.5 As2O3 without denervation?

If there is no denervation effect, It is necessary to consider what is the reason for the difference between human and mouse.

Is there any subclinical muscle atrophy in humans?

Author Response

Reviewer#2

In this paper, authors investigated the toxic effect of As2O3 on muscle. They found that both 0.05 and 0.5 ppm As2O3 increased the fasting plasma glucose level; but only 0.5 ppm arsenic exposure significantly decreased muscle mass, muscle endurance, and cross-sectional area of muscle fibers. Their conclusion is that inorganic arsenic at doses relevant to human may possess the myoatrophic potential. Regarding the presence or absence of muscular atrophy of arsenic, there is no previous report and it is a new focus. Studies have also carefully considered the cause using atrogin-1. The reviewer has only some minor concerns.

  1. Although past reports have shown that inorganic arsenic has little effect on muscle, it has a strong effect on peripheral nerves in human (Int J Mol Sci. 2019 Jul 11;20(14):3418. doi: 10.3390/ijms20143418). In histological analysis, are there grouping atrophy of muscle fibers in 0.5 As2O3 without denervation?

Response: We appreciate the reviewer's comments. We have cited this reference (Int J Mol Sci. 2019 Jul 11;20(14):3418) for arsenic neurotoxicity in humans in the text of this revised manuscript. Moreover, as shown in Figure 4, there was atrophy of muscle fibers in 0.5 ppm As2O3-treated normal mice without denervation. We have rewritten the description for this issue.

  1. If there is no denervation effect, It is necessary to consider what is the reason for the difference between human and mouse.

Response: We appreciate the reviewer's comments. We tried to respond this issue as follows. Exposure of inorganic arsenic can induce the neurotoxicity in experimental animals and humans [Refs. 13-15]. In the present study, we found that 0.5 ppm As2O3 could induce atrophy of muscle fibers in normal mice without or with denervation. Nevertheless, the human neurodegenerative disorders are known to induce muscle atrophy, such as spinal and bulbar muscular atrophy [Refs. 31,32]. The role of neurotoxicity of androgens (e.g. testosterone and its derivative dihydrotestosterone) in human spinal and bulbar muscular atrophy has been suggested [Ref. 32]. Therefore, we may not be able to rule out the role of neurotoxicity of arsenic in 0.5 ppm As2O3-induced muscle atrophy in normal mice without denervation.

We added these descriptions in the Discussion of this revised manuscript.

  1. Is there any subclinical muscle atrophy in humans?

Response: We appreciate the reviewer's comments. We tried to respond this issue as follows. There is the subclinical muscle atrophy in humans. Rocchi et al. (2011) have suggested that subclinical involvement of the autonomic dysfunction in human spinobulbar muscular atrophy [Ref. 31]. The subclinical features have been characterized in an autosomal dominant spinal muscular atrophy caused by the heterozygous variants in BICD cargo adapter 2 (BICD2) [Ref. 33]. Fischmann et al. (2012) have also shown that the subclinical changes in patients with oculopharyngeal muscular dystrophy can be detected by conventional and quantitative muscle magnetic resonance imaging (MRI) [Ref. 34]. However, the subclinical muscle atrophy induced by arsenic exposure in animals or humans still remains further investigations.

We added these descriptions in the Discussion of this revised manuscript.